# Vector competence and immune response of *Aedes aegypti* for Ebinur Lake virus, a newly classified mosquito-borne orthobunyavirus

Cihan Yang[1,2◎], Fei Wang[1◎], Doudou Huang[1], Haixia Ma[1], Lu Zhao[1], Guilin Zhang[3], Hailong Li[4], Qian Han[5], Dennis Bente[6], Ferdinand Villanueva Salazar[7], Zhiming Yuan[1,2]*, Han Xia[1,2]*

**1** Key Laboratory of Special Pathogens and Biosafety, Wuhan Institute of Virology, Chinese Academy of Sciences, Wuhan, Hubei, China, **2** University of Chinese Academy of Sciences, Beijing, China, **3** Xinjiang Heribase Biotechnology CO., LTD., Urumqi, Xinjiang, China, **4** Center for Disease Control and Prevention of Xinjiang Military Command Area, Urumqi, Xinjiang, China, **5** One Health Institute, Hainan University, Haikou, Hainan, China, **6** Department of Microbiology & Immunology, University of Texas Medical Branch, Galveston, Texas, United States of America, **7** Department of Medical Entomology, Research Institute for Tropical Medicine, Muntinlupa City, Philippines

◎ These authors contributed equally to this work.
* yzm@wh.iov.cn (ZY); hanxia@wh.iov.cn (HX)

**Data Availability Statement:** All relevant data are within the manuscript and its Supporting Information files.

## Abstract

The global impact of mosquito-borne diseases has increased significantly over recent decades. Ebinur Lake virus (EBIV), a newly classified orthobunyavirus, is reported to be highly pathogenic in adult mice. The evaluation of vector competence is essential for predicting the arbovirus transmission risk. Here, *Aedes aegypti* was applied to evaluate EBIV infection and dissemination in mosquitos. Our experiments indicated that *Ae. aegypti* had the possibility to spread EBIV (with a transmission rate of up to 11.8% at 14 days post-infection) through biting, with the highest viral dose in a single mosquito's saliva reaching 6.3 plaque-forming units. The highest infection, dissemination and ovary infection rates were 70%, 42.9%, and 29.4%, respectively. The high viral infection rates in *Ae. aegypti* ovaries imply the possibility of EBIV vertical transmission. *Ae. aegypti* was highly susceptible to intrathoracic infection and the saliva-positive rate reached 90% at 10 days post-infection. Transcriptomic analysis revealed Toll and Imd signaling pathways were implicated in the mosquito's defensive response to EBIV infection. *Defensin C* and *chitinase 10* were continuously downregulated in mosquitoes infected via intrathoracic inoculation of EBIV. Comprehensive analysis of the vector competence of *Ae. aegypti* for EBIV in laboratory has indicated the potential risk of EBIV transmission through mosquitoes. Moreover, our findings support a complex interplay between EBIV and the immune system of mosquito, which could affect its vector competence.

**Funding:** HX received the fund from Alliance of International Science Organizations (No. ANSO-CR-PP-2020-05). The funders had no role in study design, data collection and analysis, decision to publish, or preparation of the manuscript.

**Competing interests:** The authors have declared that no competing interests exist.

## Author summary

*Aedes aegypti*, the major vector of numerous clinically important arboviruses such as dengue virus, is frequently used in studies of vector competence for arboviruses. Our findings imply that *Ae. aegypti* can get infected with EBIV following either oral feeding or intrathoracic injection, posing a risk of viral transmission. After oral infection, EBIV was able to enter saliva when the viral titer in the gut exceeded a threshold. Remarkably, the saliva-positive rates in mosquitoes infected by intrathoracic injection were greater than those infected by oral feeding. The results of oral feeding and intrathoracic injection indicate that the midgut escape barrier (MEB) is the principal barrier to EBIV infection in *Ae. aegypti*. According to our analysis of the changes in gene expression after infection, we also found that the expression of some immune-related genes changed during early viral infection, whereas the expression of mosquito genes involved in metabolism showed significant differences at late stage of virus infection.

## 1 Introduction

Mosquito-borne viruses (MBV), a group of heterogeneous RNA viruses, naturally replicate in both mosquitoes and vertebrate hosts and are the etiological agents of several human and animal diseases [1]. The clinically important MBVs are primarily distributed into four families: *Flaviviridae* (dengue viruses 1–4 (DENV), Zika virus (ZIKV), West Nile virus, Japanese encephalitis virus), *Togaviridae* (Chikungunya virus), *Reoviridae* (Yunnan orbivirus) and *Peribunyaviridae* (Bunyamwera virus (BUNV)) [2–4]. Over the past few decades, the extensive global spread of arbovirus has been an issue of significant concern. In view of the dramatic emergence and unprecedented rapid spread of epidemic arboviral diseases, strong surveillance and risk assessments are necessary [5,6].

The genus *Orthobunyavirus* belonging to the family *Peribunyaviridae* (order *Bunyavirales*) contains numerous mosquito-borne bunyaviruses [7]. So far, orthobunyaviruses have been detected in various of mosquito species, including *Ochlerotatus* spp., *Culex* spp. and *Aedes* spp., and midges, such as *Culicoides paraenesis*, a vector of Oropouche virus (OROV) in South America [8–10]. A number of viruses in this genus cause severe human diseases, for instance, acute but self-limiting febrile illness (OROV), encephalitis (OROV and La Crosse virus (LACV)) and hemorrhagic fever (Ngari virus) [11,12]. Widespread epidemic distribution of orthobunyaviruses within the human population is reported, such as BUNV, which is endemic in many African countries as well as North America, and Batai virus (BATV) that is geographically prevalent in Europe [12]. In view of the global expansion of mosquito vectors, these viruses pose a considerable threat to human and animal health as well as food security.

Ebinur Lake virus (EBIV), a newly identified orthobunyavirus in China, was originally isolated from *Culex modestus* mosquito pools in Xinjiang Province [13]. The whole genome sequence of EBIV shares the greatest similarity with Germiston virus originating from South Africa [14]. EBIV has been shown to efficiently infect cell lines derived from rodent, avian, non-human primate, mosquito and human subjects [15]. Additionally, EBIV induces encephalopathy, hepatic and immunological system damage with a high mortality index in experimentally infected BALB/c mice [16]. While no confirmed human cases of EBIV have been recorded, IgM and/or IgG positive for EBIV were identified from several serum samples and two neutralizing antibody-positive cases detected via the plaque reduction neutralization test assay in an earlier study by Xia et al. [15].

*Aedes aegypti* is an important vector of MBVs [17]. Originally, *Ae. aegypti* was mainly distributed in Africa and Southeast Asia but has since colonized almost all other continents [18]. The distribution of *Ae. aegypti* is continuously expanding, with a further ~20% increase estimated by the end of this century [19]. Assessment of the vector competence of *Ae. aegypti* for transmission of newly emerging viruses is therefore critical for management of infection rates. And *Ae. aegypti* has been identified as a vector for many orthobunyaviruses, such as BUNV, LACV, and Cache Valley virus [12,20–22]. In the current study, we evaluated EBIV infection rates of *Ae. Aegypti* though both blood-feeding and intrathoracic injection routes. Transcriptome analysis of intrathoracically infected *Ae. aegypti* was additionally performed to explore the innate immune response of mosquito. The collective results provide valuable insights into the interactions between orthobunyaviruses and mosquito vectors that should aid in assessing the potential risk of EBIV.

## 2 Materials and methods

### 2.1 Viruses and cell lines

BHK cells were maintained in Dulbecco's modified Eagle's medium (Gibco) containing 10% fetal bovine serum (Bio-One) under 37°C and 5% $CO_2$. *Aedes albopictus* C6/36 cells were grown in Roswell Park Memorial Institute (RPMI) 1640 medium (Gibco) supplemented with 10% FBS (Gibco) at 28°C and 5% $CO_2$.

EBIV (strain Cu20-XJ) was stored in our laboratory. The titer for the working stock of EBIV (propagated in BHK-21 cells) was $2.4 \times 10^7$ plaque-forming units $mL^{-1}$ (PFU $mL^{-1}$). Viral titers were determined using the plaque formation assay [23].

### 2.2 Mosquito rearing

Eggs of *Ae. aegypti* (Rockefeller strain) were acquired from the Laboratory of Tropical Veterinary Medicine and Vector Biology at Hainan University. Eggs and larvae of *Ae. aegypti* were maintained in a mosquito room under the following conditions: 28°C under a light:dark cycle of 12:12 h and relative humidity of 75% ± 5%. Emerged adult mosquitoes were fed an 8% glucose solution and maintained in mesh cages (30 × 30 × 30 cm) within incubators at 28 ± 1°C under relative humidity of 80% and light:dark cycle of 12:12 h. Mosquitoes were reared in an arthropod containment level 1 (ACL-1) laboratory.

### 2.3 Oral infection and intrathoracic inoculation

Prior to oral infection and intrathoracic inoculation, female mosquitoes (5 to 8 day-old) were starved for 24 h. The mosquito infection works were performed in an ACL-2 laboratory.

**2.3.1 Infection via blood feeding.**   A virus-blood mixture (the defibrinate horse blood and virus ratio of 1:1) was used to feed female *Ae. aegypti* through an artificial mosquito feeding system (Hemotek). Fully engorged female mosquitoes (n ≥ 30) were transferred into new containers and maintained in an incubator at 28°C and humidity of 80% for 4 to 14 days until experimental use.

1. In a previous study by our group, following infection of adult female BALB/c mice with 10 PFU of EBIV, viremia reached $10^6$ PFU $ml^{-1}$ at 2 dpi [16]. Based on this finding, female mosquitoes fed a virus-blood mixture (final virus concentration of $3.7 \times 10^6$ PFU $ml^{-1}$) were used to evaluate whether EBIV could effectively infect *Ae. aegypti*. Infected mosquitoes were subsequently collected at 4, 7, 10 and 14 dpi for viral RNA determination.

2. To establish the minimum concentration of EBIV in *Ae. aegypti*, female mosquitoes were fed different virus-blood concentrations (six serial titers ranging from $10^2$ to $10^7$ PFU ml$^{-1}$). Infected mosquitoes were subsequently collected at 4 and 10 dpi for viral RNA determination.

3. To determine EBIV distribution in infected mosquitoes via artificial blood feeding (final viral titer of $6.4 \times 10^6$ PFU ml$^{-1}$), the presence of viral RNA in saliva, head, gut, and ovary of mosquitoes at 4, 7, 10 and 14 dpi was examined. For collection of mosquito saliva, a previously reported protocol was employed [24]. Firstly, the wings and legs of mosquitoes were cut off, following which mouthparts were inserted into pipette tips filled with immersion oil. Mosquitoes secreted saliva into the oil for a 45–60 min period at room temperature. Different tissues examined under a dissecting microscope.

Four parameters were calculated, specifically, infection rate (%) = $100 \times$ (number of mosquitoes with virus-positive bodies or midguts/number of total mosquitoes), dissemination rate (%) = $100 \times$ (number of mosquitoes with virus-positive heads/number of mosquitoes with virus-positive bodies or midguts), transmission rate (%) = $100 \times$ (number of mosquitoes with virus-positive saliva/number of mosquitoes with virus-positive bodies or midguts) and ovary infection rate (%) = $100 \times$ (number of mosquitoes with virus-positive ovaries/number of mosquitoes with virus-positive midguts).

**2.3.2 Infection via intrathoracic injection.** Intrathoracic injection was performed as follows: mosquitoes were anesthetized at -20˚C for 1 min. Female mosquitoes were selected and placed on an ice plate. Under the dissecting microscope, a loaded needle (filled with EBIV) was inserted into the mosquito thorax using a Nanoject III auto-nanoliter injector (Drummond). Each mosquito was administered 100 nL EBIV by pressing the INJECT button. Injected mosquitoes (n = 30) were transferred into new containers and maintained in the incubator at 28˚C and humidity of 80% for 2 to 14 days.

1. To evaluate the effects of the different barriers of *Ae. aegypti* after EBIV infection, female mosquitoes were injected with 340 PFU virus and subsequently collected at 2, 4, 7, 10 and 14 dpi for viral RNA determination.

2. To establish the minimum concentration of EBIV infection in *Ae. aegypti*, female mosquitoes were injected with different doses of virus (three serial concentrations ranging from 0.34 to 340 PFU) and infected mosquitoes collected at 7 dpi for viral RNA determination.

3. To assess EBIV distribution in infected mosquitoes subjected to intrathoracic injection (at a viral dose of 340 PFU), viral RNAs in saliva, head, gut, and ovary of mosquitoes at 2, 4, 7 and 10 dpi were determined and tissue/saliva collection was performed as above.

Four parameters were calculated, specifically, gut infection rate (%) = $100 \times$ (number of mosquitoes with virus-positive guts/number of total mosquitoes), head infection rate (%) = $100 \times$ (number of mosquitoes with virus-positive heads/number of total mosquitoes), saliva-positive rate (%) = $100 \times$ (number of mosquitoes with virus-positive saliva/number of total mosquitoes) and ovary infection rate (%) = $100 \times$ (number of mosquitoes with virus-positive ovaries/number of total mosquitoes).

## 2.4 qRT-PCR analysis of viral RNA

To detect the virus load in whole female mosquitoes or tissue/saliva samples from female mosquitoes, each mosquito or tissue/saliva sample was placed in 200–300 μL RPMI 1640 and stored at -80˚C until experimental use. All samples were initially homogenized using a Low

Temperature Tissue Homogenizer Grinding Machine (Servicebio) (operating frequency = 60 Hz, operation time = 15 s, pause time = 10 s, cycles = 2, and setting temperature = 4˚C), followed by centrifugation for 10 min at 12, 000 × g min and 4˚C. Total RNA of each sample was extracted using an automated nucleic acid extraction system following the manufacturer's instructions (NanoMagBio).

Using the CFX96 Touch Real-Time PCR Detection System (Bio-Rad) and One Step TB Green PrimeScript PLUS RT-PCR Kit (Takara), viral RNA copies of each sample were quantified. The primers used are presented in S1 Table. The cutoff for EBIV-positive samples determined via qRT-PCR was Ct < 35. The positive cutoff value was evaluated by comparing paired serial ten-fold dilutions either inoculated on cells or assayed via qRT-PCR (S2 Table) [25]. The equation for the standard curve was $y = -3.5566x + 37.887$ [$x$ = lg (titer of EBIV), $y$ = Ct value and $R^2$ = 0.9979], which was generated using 10-fold serial dilutions of virus ($1.3 \times 10^6$ PFU ml$^{-1}$) and used to calculate the EBIV titer in each sample [26].

## 2.5 Immunohistochemical detection of EBIV antigen

Midguts and ovaries were dissected from orally infected female mosquitoes (n = 30) at 14 dpi and intrathoracically infected female mosquitoes (n = 15) at 7 dpi, respectively. Tissues were fixed using 4% paraformaldehyde for 1 h and washed with PBS containing 0.3% Triton X-100 (PBST) five times. Next, tissues were placed in blocking solution (PBS containing 5% goat serum and 0.3% Triton X-100) for 1 h and incubated with primary mouse anti-EBIV-NP antibody (derived from mice serum, diluted 1:200 in PBST containing 5% goat serum) for 24 h, followed by secondary Alexa Fluor 549-conjugated goat anti-mouse IgG (H+L) (diluted 1:250 in PBST containing 5% goat serum; Invitrogen) overnight. The actin cytoskeleton was stained with Alexa Fluor 488 Phalloidin (Invitrogen) for 1 h. After each step, tissues were washed at least five times in 0.3% PBST to avoid effects of reagents on subsequent affecting following operations. Finally, tissues were mounted onto slides using SlowFade Diamond Antifade Mountant (Invitrogen) and images recorded with the aid of a Leica SP8 confocal microscope (filter information TD 458/514/561; Leica, Germany). Using LAS X software (Leica), z-stack images were merged and scale bars added. PowerPoint 2019 was utilized for image grouping. All samples were analyzed under the same microscope and software settings.

## 2.6 Visualization of EBIV particles via transmission electron microscopy

Ovaries and midguts were obtained from female mosquitoes (n = 20) infected via intrathoracic inoculation at 7 dpi with the aid of a dissection microscope and tissues fixed in 2.5% glutaraldehyde until experimental use. Fixed samples were handled in the Center for Instrumental Analysis and Metrology (Wuhan Institute of Virology, China), sectioned using an ultramicrotome and visualized under a Tecnai G20 TWIN transmission electron microscope (FEI, United States). PowerPoint 2019 was used for image grouping.

## 2.7 Transcriptomic analysis

Female mosquitoes infected via intrathoracic inoculation and mock-infected mosquitoes were collected at 2 and 7 dpi. Each pool (comprising ten mosquitoes) was subjected to total RNA extraction using TRIzol reagent (Invitrogen). Three independent biological replicates were prepared for each sample. Next, samples were delivered to Wuhan Benagen Tech Solutions Company for commercial RNA-seq and data analysis. RNA degradation and contamination were monitored via agarose gel electrophoresis on 1% gels. RNA purity was assessed using the NanoPhotometer spectrophotometer (IMPLEN, CA, USA) and RNA integrity determined using the RNA Nano 6000 Assay kit of the Bioanalyzer 2100 system (Agilent Technologies,

CA, USA). Sequencing libraries were generated using NEBNext Ultra RNA Library Prep Kit for Illumina (NEB, USA) following the manufacturer's recommendations and index codes added to attribute sequences to each sample. Clustering of index-coded samples was performed on a cBot Cluster Generation System using the TruSeq PE Cluster Kit v3-cBot-HS (Illumina) according to the manufacturer's instructions. After cluster generation, library preparations were sequenced on an Illumina Novaseq platform and 150 bp paired-end reads generated. Clean reads were obtained by removing reads containing adapter, reads containing ploy-N and low-quality reads from raw data and subsequently used for *de novo* assembly using Trinity program (http://trinityrnaseq.sourceforge.net/) and mapped to the *Ae. aegypti* genome database (RefSeq: GCF_002204515.2). The Unigene sequences of samples were searched against the Nr, KEGG and GO databases (E-value ≤ 1E-5) using BLASTX to retrieve protein functional annotations based on sequence similarity. The fragments per kilobase of exon per million mapped fragments values were directly applied to compare gene expression differences between samples. The DESeq package was employed to obtain the "base mean" value for identification of differentially expressed genes (DEGs). Absolute value of $\log_2$ ratio ≥ 1 and $p ≤ 0.05$ were set as the thresholds for significance of gene expression differences between two samples. Scatter diagrams and bubble diagrams were generated with GraphPad Prism statistical software 9.1.0.

## 2.8 Statistical analysis

All data were analyzed with GraphPad Prism statistical software 9.1.0. Differences in continuous variables and mosquito infection, dissemination, transmission and ovary infection rates were analyzed using the non-parametric Kruskal-Wallis test for multiple comparisons and Fisher's exact test where appropriate, as specified in the figure legends. *P* values ≤ 0.05 were considered statistically significant.

## 3. Results

### 3.1 EBIV is disseminated from midgut to saliva in *Ae. aegypti* through blood feeding

The mean viral titers of EBIV-positive mosquitoes at the four time-points were not significantly different and all values were above $10^2$ PFU ml$^{-1}$. Notably, viral levels in some infected mosquitoes were remarkably higher, with the highest viral titer recorded as $10^{6.4}$ PFU ml$^{-1}$ (Fig 1A). The infection rates at the four time-points ranged from 40.3% to 70.7%. The infection rate at 4 dpi was significantly higher than that at 7 ($p = 0.0086$) and 14 dpi ($p = 0.0005$) and comparable with that at 10 dpi (Fig 1B).

With increasing levels of EBIV in blood meals, the proportion of infected mosquitoes gradually increased and mean viral titers at 4 dpi and 10 dpi in mosquitoes fed the same dose of EBIV showed no significant differences (Fig 1C). The infection rates of mosquitoes fed $10^6$ and $10^7$ PFU ml$^{-1}$ EBIV at 4 dpi (30.3% and 62.2%) and 10 dpi (48.6% and 52.4%) were significantly higher relative to the other groups while differences between these groups were not marked (Fig 1D).

The mean titers in EBIV-positive guts at 4 ($10^{2.7}$ PFU mL$^{-1}$), 7 ($10^{2.9}$ PFU mL$^{-1}$), and 10 ($10^{2.8}$ PFU mL$^{-1}$) dpi were higher than those in heads, saliva and ovaries (Fig 1E). At 14 dpi, the mean viral titer in heads ($10^{3.5}$ PFU mL$^{-1}$) was comparable to that in guts ($10^{3.2}$ PFU mL$^{-1}$) (Fig 1F). Notably, the mean viral titer in saliva was highest at 14 dpi ($10^{2.5}$ PFU mL$^{-1}$) (Fig 1G) and at this time-point, the highest mean virus titer was also detected in the ovary ($10^{3.6}$ PFU mL$^{-1}$) (Fig 1H). The infection rates at the four time-points ranged from 50%-70% and were

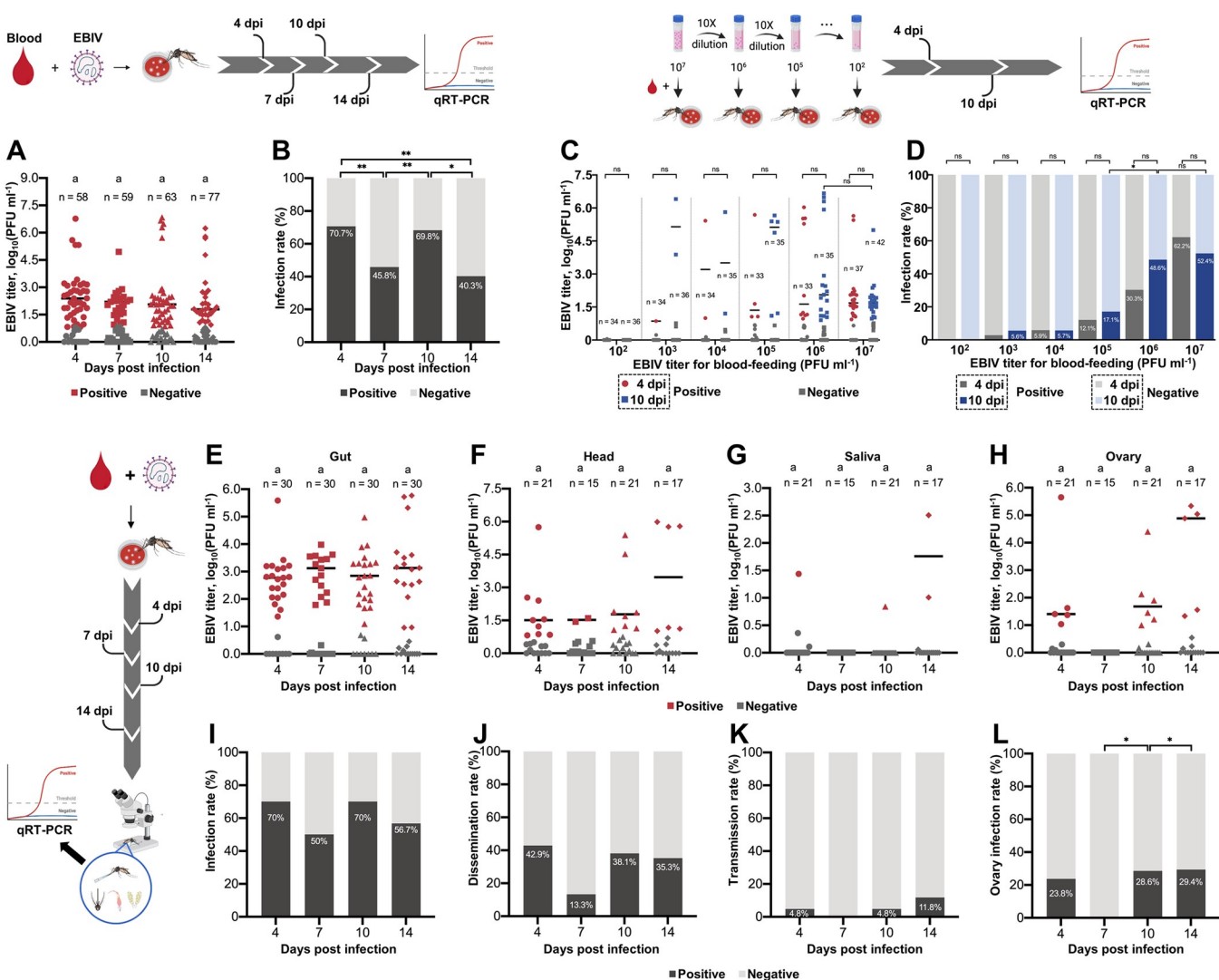

**Fig 1. EBIV infection rates of *Ae. aegypti* through oral feeding.** EBIV titers (A) and infection rates (B) of mosquitoes at 4, 7, 10 and 14 days after feeding on blood meal containing $3.7 \times 10^6$ PFU ml$^{-1}$ EBIV. EBIV titers (C) and infection rates (D) of mosquitoes from six serial viral titer groups at 4 and 10 days after feeding on blood meal containing $10^2$ to $10^7$ PFU ml$^{-1}$ EBIV. EBIV titers in gut (E), head (F), saliva (G) and ovary (H) samples of mosquitoes at 4, 7, 10 and 14 days after feeding on blood meal containing $6.4 \times 10^6$ PFU ml$^{-1}$ EBIV. Infection rates (I), dissemination rates (J), transmission rates (K) and ovary infection rates (L) of mosquitoes at 4, 7, 10 and 14 days after feeding on blood meal containing $6.4 \times 10^6$ PFU ml$^{-1}$ EBIV. Each dot represents an individual mosquito. The same letters indicate no significant differences (multiple comparisons using non-parametric Kruskal-Wallis analysis). Differences in the rates were analyzed with Fisher's exact test (\*: $p \le 0.05$, \*\*: $p \le 0.01$, \*\*\*: $p \le 0.005$ and \*\*\*\*: $p \le 0.001$).

not significantly different (Fig 1I). The dissemination rates ranged from 13.3% to 42.9% (Fig 1J). The transmission rates ranged from 0% to 11.8%, with the highest rate (11.8%) recorded at 14 dpi (Fig 1K). The ovary infection rates at 4, 10 and 14 dpi were 23.8%, 28.6% and 29.4%, respectively. All ovary samples at 7 dpi did not contain detectable levels of virus (Fig 1L).

### 3.2 *Ae. aegypti* is highly susceptible to EBIV infection through intrathoracic inoculation

After injection of 340 PFU EBIV, mean viral titer of EBIV-positive mosquitoes at 2 dpi ($10^{5.5}$ PFU mL$^{-1}$, H = 48.01, $p < 0.0001$) was significantly higher than that at 4, 7 and 10 dpi. The

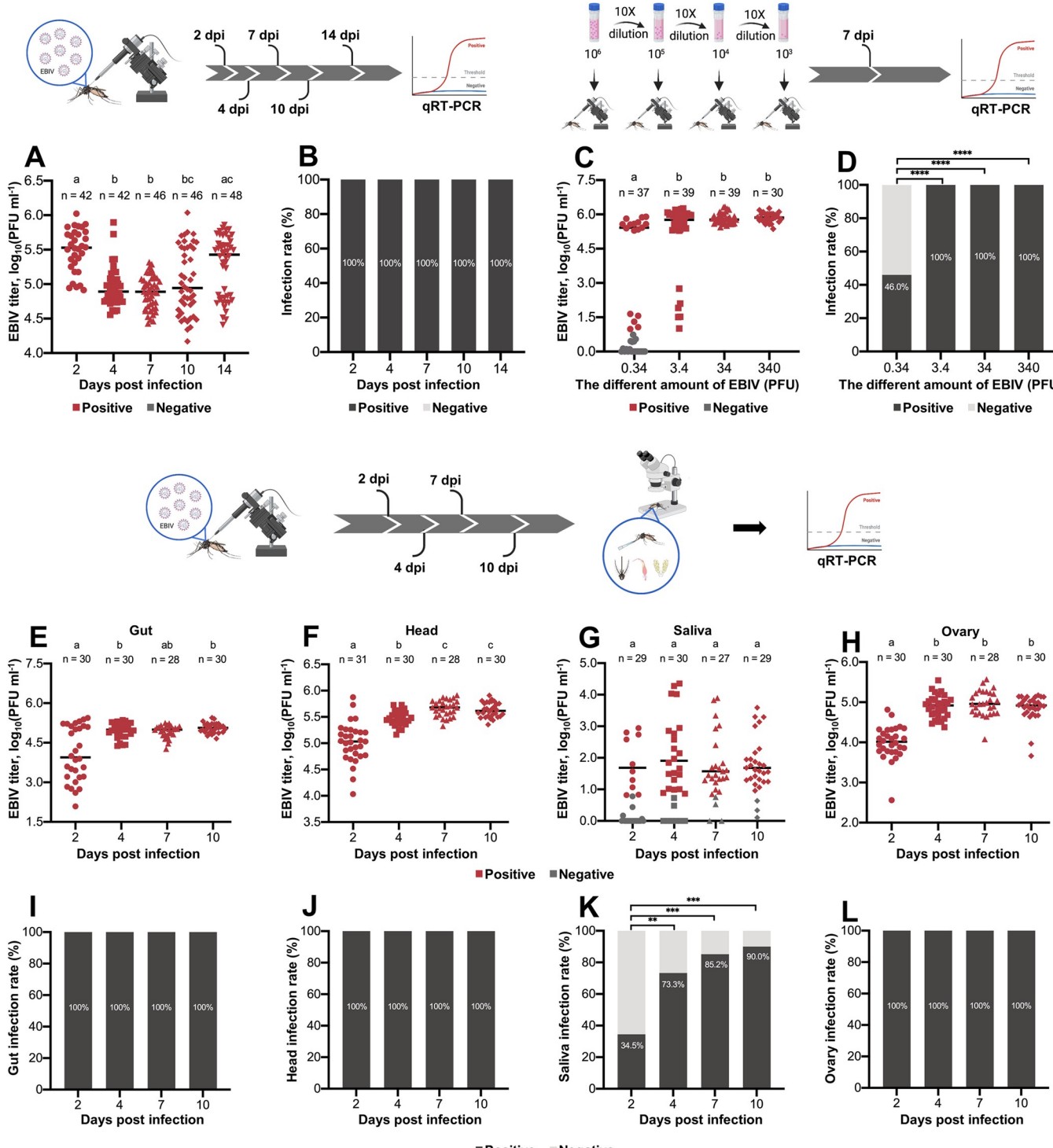

**Fig 2. EBIV titers and infection rates of *Ae. aegypti* through intrathoracic inoculation.** EBIV titers (A) and infection rates (B) of mosquitoes at 4, 7, 10 and 14 days after injection with 340 PFU EBIV. EBIV titers (C) and infection rates (D) of mosquitoes injected with 0.34 to 340 PFU EBIV at 7 dpi. EBIV titers in gut (E), head (F), saliva (G) and ovary (H) samples of mosquitoes injected with 34 PFU EBIV at 2, 4, 7 and 10 dpi. Gut (I), head (J), saliva (K) and ovary (L) infection rates of mosquitoes injected with 34 PFU EBIV at 2, 4, 7 and 10 dpi. Each dot represents an individual mosquito. The same letters indicate no significant differences (multiple comparisons using non-parametric Kruskal-Wallis analysis). Differences in rates were analyzed with Fisher's exact test (*: $p \leq 0.05$, **: $p \leq 0.01$, ***: $p \leq 0.005$ and ****: $p \leq 0.001$).

mean viral titer at 14 dpi ($10^{5.3}$ PFU ml$^{-1}$) was the second highest among all five time-points (Fig 2A). Infection rates at the five time-points were 100% (Fig 2B).

The mean viral titer was the lowest in EBIV-positive mosquitoes injected with 0.34 PFU EBIV ($10^{4.3}$ PFU ml$^{-1}$). Mean viral titer values of EBIV-positive mosquitoes injected with 3.4, 34 and 340 PFU EBIV were $10^{5.1}$ PFU mL$^{-1}$, $10^{5.8}$ PFU mL$^{-1}$ and $10^{5.9}$ PFU mL$^{-1}$, respectively, which were not significantly different (Fig 2C). The infection rates of all groups of mosquitoes were 100%, except that injected with 0.34 PFU EBIV, which had an infection rate of only 46% (Fig 2D).

Mean titers in EBIV-positive guts at 4 dpi ($10^{5.0}$ PFU ml$^{-1}$) and 10 dpi ($10^{5.1}$ PFU mL$^{-1}$) were significantly higher than that at 2 dpi ($10^{4.1}$ PFU ml$^{-1}$, H = 13.70, $p$ = 0.0033) (Fig 2E). The mean titers in EBIV-positive head samples were gradually increased at the four time-points, with the highest value of $10^{5.7}$ PFU ml$^{-1}$ observed at 7 dpi (H = 63.88, $p$ < 0.0001) (Fig 2F). Notably, viral titers in saliva at the four time-points were not significantly different and the highest recorded value was $10^{2.2}$ PFU ml$^{-1}$ at 4 dpi (H = 20.27 and $p$ = 0.0001) (Fig 2G). Mean viral titers in ovary samples were also high at 4 dpi ($10^{4.9}$ PFU ml$^{-1}$), 7 dpi ($10^{5.0}$ PFU ml$^{-1}$) and 10 dpi ($10^{4.9}$ PFU ml$^{-1}$) relative to that at 2 dpi ($10^{4.0}$ PFU ml$^{-1}$) (Fig 2H). Gut, head and ovary infection rates were 100% (Fig 2I, 2J and 2L). With increasing days after infection, saliva infection rates were slightly increased, ranging from 34.5% to 90%. The saliva infection rate was highest (up to 90%) at 10 dpi ($p$ < 0.0001, comparison between 2 and 10 dpi) (Fig 2K).

### 3.3 Presence of EBIV antigens and viral particles in infected mosquitoes

Immunohistochemical analysis confirmed the presence of EBIV antigen in the midgut region of orally infected mosquitoes at 14 dpi (Fig 3A) and midgut and ovary samples of mosquitoes infected via intrathoracic inoculation at 7 dpi (Fig 3B and 3C). The EBIV antigen accumulated around the cytoskeleton in midgut endothelial cells (Fig 3A and 3B). In ovarioles, viral signals accumulated around the nucleus in nurse cells but not follicle cells (Fig 3C). The EBIV antigen was not detected in tissues from uninfected mosquitoes (Fig 3D–3F).

To confirm the presence of EBIV viral particles in infected mosquitoes, sections of the digestive tract and ovaries from mosquitoes infected via intrathoracic inoculation were prepared and examined via transmission electron microscopy. Notably, spherically shaped virus-like particles (VLPs) 78–140 nm in diameter were stacked in intracellular vesicles and detectable in cells of the digestive tract (Fig 3G–3J). Similarly, VLPs 60–100 nm in diameter were present in nurse cells of the ovary (Fig 3K–3N). As expected, VLPs were not detectable in uninfected mosquitoes.

### 3.4 Immune and metabolic responses of *Ae. aegypti* to EBIV infection

Transcriptome experiments showed significant upregulation of 17 genes and downregulation of 28 genes in mosquitoes subjected to intrathoracic inoculation of EBIV at 2 dpi while significant upregulation of 45 genes and downregulation of 65 genes was observed in infected mosquitoes at 7 dpi ($p$ value ≤ 0.05, |log2foldchange| ≥ 1) (Fig 4A and 4B).

Limited genes were upregulated in mosquitoes infected via intrathoracic inoculation at 2 dpi, most of which were undefined (S3 Table). The top three upregulated genes encoded macrophage erythroblast attacher-like (E3 Ubiquitin Ligase), zinc finger protein 37 homolog isoform X1/X2 and beta-galactoside-binding lectin-like (Fig 4A). Among the upregulated genes at this time-point, adult-specific cuticular protein 20 (*ACP-20*) showed the highest expression (S3 Table). Moreover, the number of downregulated genes was similar with that of upregulated genes. The most of proteins encoded by these genes were related to cellular immunity and

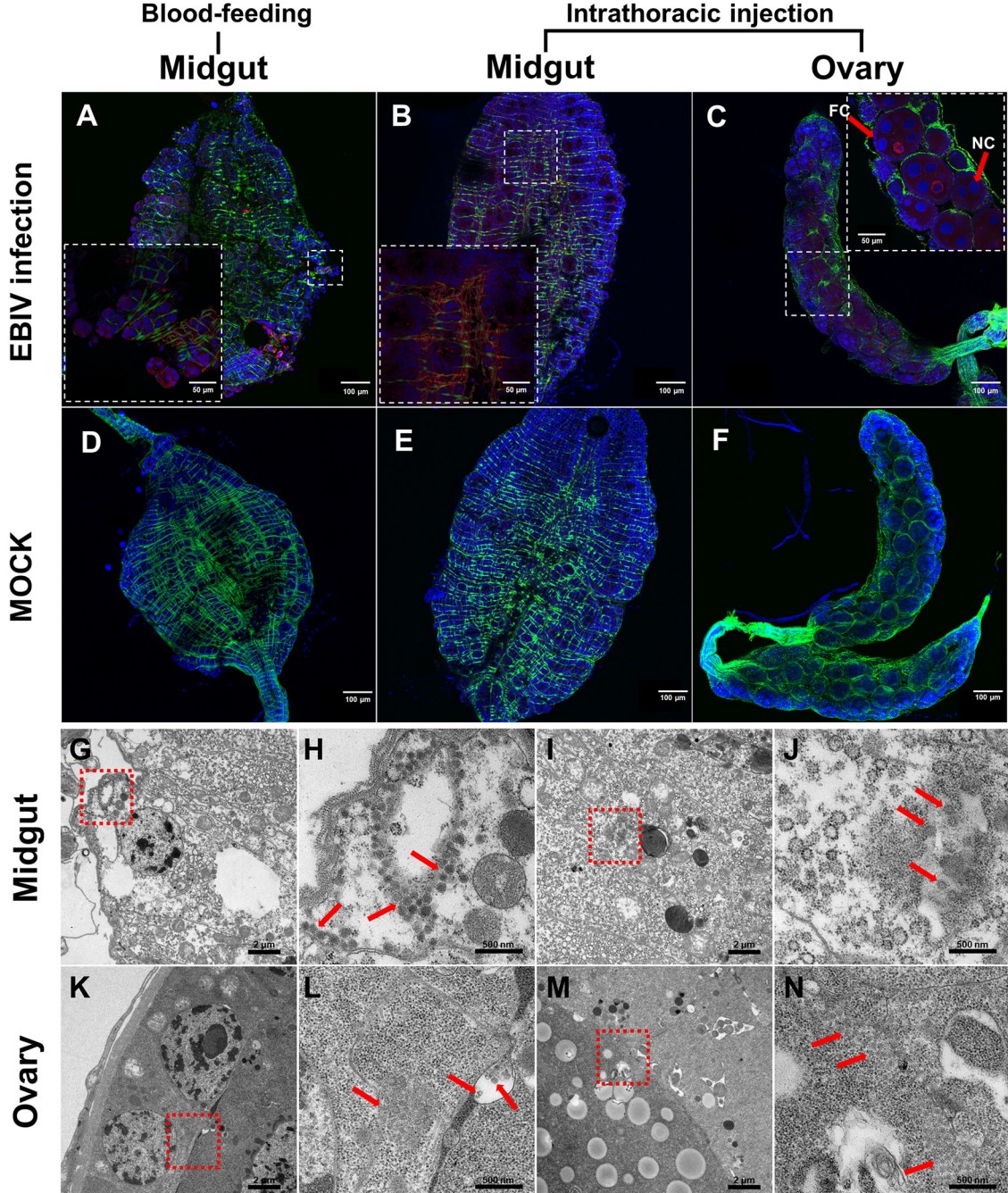

**Fig 3. Immunohistochemical visualization and electron micrographs of EBIV in *Ae. aegypti* midgut and ovary.**
Immunolocalization of EBIV antigen in midguts of mosquitoes with oral infection at 14 dpi (A) and intrathoracic inoculation at 7 dpi (B). Immunolocalization of EBIV antigen in ovaries of mosquitoes infected via intrathoracic inoculation at 7 dpi (C). Immunolocalization of EBIV antigen in midguts of mosquitoes fed blood without EBIV at 14 dpi (D) and mosquitoes injected with 100 nl RPMI 1640 at 7 dpi (E). Immunolocalization of EBIV antigen in ovaries from mock-injected mosquitoes at 7 dpi (F). F-actin was stained with phalloidin (green). The cell nucleus was stained with DAPI (blue). NC: nurse cells, FC: follicle cells. (G-J) Virions observed in gut are indicated by red arrows on electron micrographs. H and J are the enlarged insets of the boxes in G and I, respectively. (K-N) Virions observed in ovary are indicated by red arrows on electron micrographs. L and N are the enlarged insets of the boxes in K and M, respectively. EBIV virion clusters were detected using a mouse anti-EBIV polyclonal antibody and goat anti-mouse IgG labeled with red fluorescent secondary antibody.

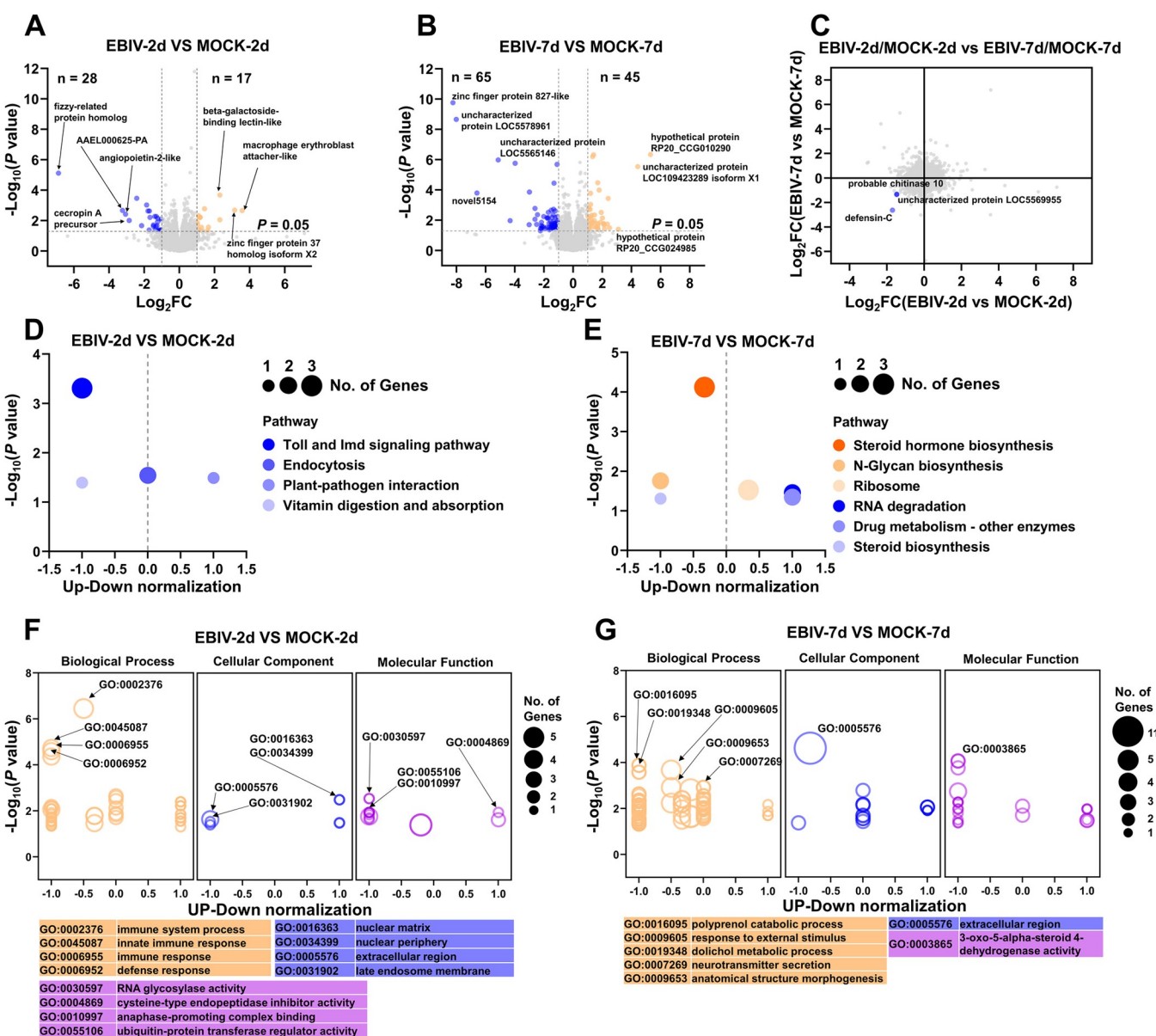

**Fig 4. EBIV affects specific gene expression patterns in intrathoracically infected mosquitoes.** Significantly upregulated (orange) and downregulated (blue) genes in EBIV-infected compared with mock-infected mosquitoes at 2 dpi (A) and 7 dpi (B). (C) Genes displaying differential responses in EBIV-infected mosquitoes at 2 and 7 dpi were determined via a scatter plot of expression changes. KEGG pathway analyses of DEGs at 2 dpi and 7 dpi are presented in (D) and (E), respectively. Enriched GO categories associated with DEGs at 2 dpi and 7 dpi are shown in (F) and (G), respectively.

apoptosis, including antimicrobial peptides (AAEL000625-PA, cecropin A precursor and defensin-C), toll-like receptor 13, cell death abnormality protein 1 isoform X2 and putative lysozyme-like protein. In addition, three odorant binding genes (*gene-LOC5577119*, *gene-LOC5571968* and *gene-LOC5567749*) and three serine protease easter genes (*gene-LOC5579410*, *gene-LOC5568135* and *gene-LOC5569890*) were significantly downregulated (S3 Table). The top four proteins encoded by downregulated genes were fizzy-related protein homolog involved in the protein ubiquitination pathway, AAEL000625-PA, angiopoietin-2-like (a growth factor whose activities are mediated through tyrosine kinase receptors [27])

and cecropin A precursor (Fig 4A). Among the downregulated genes at this time-point, *defensin-C* gene showed the greatest decrease expression (S3 Table).

Nearly half the upregulated genes in mosquitoes infected via intrathoracic inoculation at 7 dpi were undefined (S4 Table). Moreover, the top three proteins encoded by upregulated genes were uncharacterized (Fig 4B). Among the downregulated genes at this time point, *gene-LOC110676601* encoding the voltage-dependent anion-selective channel, a pore-forming protein located in the outer mitochondrial membrane, showed the greatest decrease expression (S4 Table). Among the four genes showing the most significant downregulation, the protein encoded by *gene-LOC110680747* was identified as zinc finger protein 827-like while the other three were unknown (Fig 4B). In addition, four venom allergen 5 genes (*gene-LOC5575393*, *gene-LOC5575399*, *gene-LOC5575400* and *gene-LOC5575401*) were significantly downregulated. Previous findings suggest that venom allergen 5 is associated with deltamethrin resistance in mosquitoes [28] and *Ae. aegypti* venom allergen-1 promotes DENV and ZIKV transmission through activating autophagy in host immune cells [29]. Statistical comparison of differentially expressed genes between mosquitoes infected via intrathoracic inoculation at 2 dpi and 7 dpi revealed significant differences among three genes, specifically, *defensin C*, *chitinase 10* (*gene-LOC5570579*) and *gene-LOC5569955* (Fig 4C).

KEGG pathway and GO enrichment analyses were performed to identify the biological functions and pathways activated in EBIV-infected *Ae. aegypti* at 2 and 7 dpi. KEGG results revealed significantly enriched genes in four pathways at 2 dpi, including Toll and Imd signaling and endocytosis (Fig 4D). However, the enrichment pathways at 7 dpi were mainly related to biosynthesis and metabolism, including steroid hormone biosynthesis, N-glycan biosynthesis, ribosome, RNA degradation, and steroid biosynthesis at 7 dpi (Fig 4E). Similarly, GO analysis showed significant enrichment of immune-related processes at 2 dpi and metabolism-related processes at 7 dpi (Fig 4F and 4G).

## 4. Discussion

This study explored the vector competence of *Ae. aegypti* for EBIV in view of the susceptibility of this mosquito species to other orthobunyaviruses [22,24]. For instance, *Cx. quinquefasciatus* is reported to be refractory to BUNV, but not *Ae. aegypti* [21]. Further studies on additional species and geographic strains of mosquitoes, such as *Ae. albopictus*, *Cx. quinquefasciatus, and Cx. tritaeniorhynchus*, should be conducted to identify the primary mosquito vectors for EBIV.

Generally, adult mice show resistance to orthobunyavirus infections while three-week-old or younger mice are more susceptible [16,30–32]. However, a previous study by our group showed that > 90% adult BALB/c mice succumbed to death upon intraperitoneal infection with an extremely low dose of EBIV (1–10 PFU), indicating greater pathogenicity of EBIV to mice compared to other orthobunyavirus [16]. Here, the mean viral titer of EBIV-positive saliva was > $10^{1.5}$ PFU ml$^{-1}$ (the average dose of EBIV per mosquito was > 6.3 PFU) at 14 dpi in mosquitoes infected via blood feeding, highlighting the potential risk of EBIV transmission to vertebrate hosts by mosquito. However, the issue of whether EBIV-positive mosquitoes are capable of transmitting viruses to naïve vertebrate hosts via biting is yet to be established and further studies on the EBIV-mosquito-vertebrate transmission cycle are warranted.

The highest saliva-positive rate in intrathoracic-injected mosquitoes reached 90%, compared to mosquitoes subjected to oral feeding (11.8%). Notably, after oral infection, a few EBIV-positive saliva samples of infected mosquitoes with viral gut loads > $10^4$ PFU mL$^{-1}$ were detected (Fig 1G), suggesting that EBIV can enter saliva when virus titer in the gut reaches a threshold of $10^4$ PFU ml$^{-1}$. Following intrathoracic injection, EBIV bypassed the midgut barrier and was able to rapidly infect the whole mosquito body, including saliva and ovary (Fig

2G–2H). The distinct results obtained with the two routes indicate that the midgut of *Ae. aegypti* is the main barrier to EBIV transmission. In view of the characteristics of arbovirus, which need to maintain their life cycle by switching between vertebrate hosts and invertebrate vectors, a strong possibility of evolutionary adaptation during this process is suggested [33,34]. For instance, a single amino acid substitution could increase susceptibility of *Ae. albopictus* and lead to virus dissemination more rapidly from the midgut to secondary tissues [35]. A similar situation was observed with ZIKV, whereby the mutation enhanced ZIKV infectivity in mosquitoes [36]. Therefore, once the virus evolves to increase ease of cross through the midgut barrier, it may present an emerging threat to human or animal health. In addition, it is not uncommon for orthobunyaviruses to spread during the immature life stages of mosquitoes through a mode of vertical transmission [37,38]. The high EBIV infection rates in ovary observed in the current study support the possibility of vertical transmission (Figs 1H and 2H).

During the process of invasion, arboviruses need to cope with innate immune responses and overcome several barriers of mosquito vectors. To date, four major barriers have been identified, specifically, the midgut infection barrier, midgut escape barrier (MEB), salivary gland infection barrier, and salivary gland escape barrier [39]. *Ae. aegypti* was moderately susceptible to EBIV after oral feeding, with 70.0% mosquitoes showing midgut infection at 10 dpi, whereas the dissemination rate was significantly lower than the infection rate (38.1%) at this time-point ($p = 0.0432$). Notably, the proportion of EBIV-positive saliva samples was 90.0% at 10 days post intrathoracic injection. The combined results of blood feeding and intrathoracic injection experiments suggest that MEB presents the primary barrier to systematic EBIV infection of *Ae. aegypti*. In the presence of an efficient MEB, virus replication is limited to the midgut or inefficient dissemination occurs [40]. The basal lamina of the midgut may exert effects on effective dissemination [41]. Several studies have reported an inverse relationship between LACV dissemination rates and thickness of basal lamina of *Ae. triseriatus* [42,43]. An alternative reason for ineffective dissemination could be that EBIV is unable to overcome or evade antiviral immune responses in midgut. Electron microscopy of the midgut section of mosquitoes infected via intrathoracic injection revealed the presence of a large number of autophagosomes and lysosomes, suggesting the involvement of autophagy in EBIV infection. Recent studies on arbovirus infection have demonstrated that autophagy serves as an antiviral defense mechanism [44–46]. Similarly, replication of Rift Valley fever virus belonging to the family *Phenuiviridae* is reported to be limited by activation of autophagy in *Drosophila* [47]. However, other studies suggest that autophagy is beneficial for flavivirus replication and transmission in *Ae. aegypti* [29,48]. Additionally, evidence supporting dose-dependent competence of midgut infection has been obtained [39,49,50], which indicates that with increasing viral titers in host blood, the risk of EBIV transmission to vertebrates is bigger.

Based on transcriptome analysis of intrathoracic-infected *Ae. aegypti*, Toll and Imd signaling pathways are implicated in the protective response of mosquito to EBIV infection [22]. Endocytosis plays an important role in host entry of several virus types, such as flaviviruses and bunyaviruses [51,52]. Several immune-related genes have been identified in EBIV-infected *Ae. aegypti*, such as E3 ubiquitin ligase related to autophagy [53], zinc finger protein 37 homolog isoform X1/X2 (a transcription factor) and beta-galactoside-binding lectin-like involved in the process of virus infection [54] (Fig 4A). Moreover, five zinc-finger protein-encoding genes showed differential expression patterns (*gene-LOC110680747*, *gene-LOC5571062*, *gene-LOC5576563* and *gene-LOC110680309* were downregulated while *gene-LOC5573104* was upregulated) (S4 Table). Zinc-finger proteins are one of the most abundant groups of proteins with a wide range of molecular functions. Recently, the zinc-finger protein ZFP36L1 was shown to

enhance host anti-viral defense against influenza A virus [55], supporting a role of these proteins in response to viral infection.

In conclusion, *Ae. aegypti* can be infected by EBIV and presents a potential risk of virus transmission. Further research on the vector competence of various mosquito species for this newly classified orthobunyavirus and epidemiological studies are essential to provide valuable insights that could aid in management of potential future outbreaks.

## Supporting information

**S1 Table. Primer sequences of genes used for qRT-PCR.**
(DOCX)

**S2 Table. Correlations between the EBIV-induced cytopathic effect in BHK-21 cells and CT values of virus RNA determined via qRT-PCR.**
(DOCX)

**S3 Table. Differentially expressed genes in mosquitoes infected via intrathoracic inoculation compared to mock-infected mosquitoes at 2 dpi.**
(XLSX)

**S4 Table. Differentially expressed genes in mosquitoes infected via intrathoracic inoculation compared to mock-infected mosquitoes at 7 dpi.**
(XLSX)

## Acknowledgments

We are grateful to staff at the Center for Instrumental Analysis and Metrology of Wuhan Institute of Virology, Chinese Academy of Sciences, and the Analysis and Testing Center of Institute of Hydrobiology, Chinese Academy of Sciences, for their assistance.

## Author Contributions

**Conceptualization:** Zhiming Yuan, Han Xia.

**Funding acquisition:** Han Xia.

**Investigation:** Cihan Yang, Fei Wang.

**Resources:** Doudou Huang, Haixia Ma, Guilin Zhang, Hailong Li.

**Validation:** Zhiming Yuan, Han Xia.

**Visualization:** Cihan Yang, Fei Wang.

**Writing – original draft:** Cihan Yang.

**Writing – review & editing:** Cihan Yang, Fei Wang, Lu Zhao, Guilin Zhang, Qian Han, Dennis Bente, Ferdinand Villanueva Salazar, Zhiming Yuan, Han Xia.

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
