## [Decision Letter · Decision Letter 0]

29 Mar 2022

Dear Dr. Xia,

Thank you very much for submitting your manuscript "Vector competence and transcriptional response of Aedes aegypti for Ebinur Lake virus, a newly classified mosquito-borne orthobunyavirus" for consideration at PLOS Neglected Tropical Diseases. As with all papers reviewed by the journal, your manuscript was reviewed by members of the editorial board and by several independent reviewers. In light of the reviews (below this email), we would like to invite the resubmission of a significantly-revised version that takes into account the reviewers' comments. 

We cannot make any decision about publication until we have seen the revised manuscript and your response to the reviewers' comments. Your revised manuscript is also likely to be sent to reviewers for further evaluation.

Sincerely,

Adly M.M. Abd-Alla, Prof asso.

Associate Editor

Eric Dumonteil

Deputy Editor

Reviewer's Responses to Questions

**Key Review Criteria Required for Acceptance?**

**Methods**

-Are the objectives of the study clearly articulated with a clear testable hypothesis stated?

-Is the study design appropriate to address the stated objectives?

-Is the population clearly described and appropriate for the hypothesis being tested?

-Is the sample size sufficient to ensure adequate power to address the hypothesis being tested?

-Were correct statistical analysis used to support conclusions?

-Are there concerns about ethical or regulatory requirements being met?

Reviewer #1: -Are the objectives of the study clearly articulated with a clear testable hypothesis stated? Yes

-Is the study design appropriate to address the stated objectives? Yes

-Is the population clearly described and appropriate for the hypothesis being tested? Yes

-Is the sample size sufficient to ensure adequate power to address the hypothesis being tested? Yes

-Were correct statistical analysis used to support conclusions? Yes

Reviewer #2: Methods section has a poor english. Should be entirely reviewed. 

If the objective of the study is additionally to show vector competence, than transmission to a host (animal model) should be included.

Reviewer #3: I have only a few major points for the authors to consider/clarify and a number of minor points that are easy to fix. Please see the "Summary and General Comments" section.

**Results**

-Does the analysis presented match the analysis plan?

-Are the results clearly and completely presented?

-Are the figures (Tables, Images) of sufficient quality for clarity?

Reviewer #1: -Does the analysis presented match the analysis plan? Yes

-Are the results clearly and completely presented? Yes

-Are the figures (Tables, Images) of sufficient quality for clarity? Yes

Reviewer #2: results either needs a extensive review.

Reviewer #3: I have only a few major points for the authors to consider/clarify and a number of minor points that are easy to fix. Please see the "Summary and General Comments" section.

**Conclusions**

-Are the conclusions supported by the data presented?

-Are the limitations of analysis clearly described?

-Do the authors discuss how these data can be helpful to advance our understanding of the topic under study?

-Is public health relevance addressed?

Reviewer #1: -Are the conclusions supported by the data presented? Yes

-Are the limitations of analysis clearly described? No

-Do the authors discuss how these data can be helpful to advance our understanding of the topic under study? Yes

-Is public health relevance addressed? Yes

Reviewer #2: transmission is not an adequate term to use in this manuscript, reasons pointed above. Mainly, no host was exposed to these experimentally infected mosquitoes. 

Therefore, I strongly suggest that the term "vector competence" should be changed by "vector capacity" in the manuscript in the current format. Competence should be proved by transmission experiments. Here, the virus was detected to be replicating in mosquito tissues, and viral titers were demonstrated in their saliva, but transmission experiments are lacking.

Reviewer #3: I have only a few major points for the authors to consider/clarify and a number of minor points that are easy to fix. Please see the "Summary and General Comments" section.

**Editorial and Data Presentation Modifications?**

Reviewer #1: (No Response)

Reviewer #2: This manuscript describes the experimental infection of aedes aegypti colonies by oral route and after intrathoracic administration of EBIV, a novel orthobunyavirus discovered on China. It did not report any transmission experiment, so results should be carefully reviewed since mice were not exposed to these experimentally infected mosquitoes. Despite the low viral titers detected in orally infected mosquitoes, the presence of the virus in the ovaries is of biological importance, because it can mean that this virus potentially can be transmitted vertically. To assess this information, male/female prole originated from these experimentally infected mosquitoes should be tested, since more than 20% infection rate is likely to represent this possibility of vertical transmission. Figures have na excelent quality, except for figure 4, which in this format is not possible to read (font size is to small and lacking resolution quality). The most interesting findings however, are the transvcriptome results, since studies evaluating mosquito innate response (related to apoptosis induction, antimicrobial response and ubiquitin pathway) are lacking in literature. Writting of this parto f results, and discussion section, is a better english than the rest of the manuscript.

The entire manuscript needs a english review by a english native speaker, several sentences in the abstract, introduction, material and methods and results, present gramar mistakes (plural/singular mistakes, verb conjugated in the present or future, were it should be presented in the past tense, for instance). Introduction should be more direct and concise to reflect the article content solely. Sentences are in general disconected from each other, they need a conection to let the text more “fluid”, direct, concise. Paragraphs are to big, descriptions should be re-written. Overall, the manuscript should be reviewed to remove non necessary and repetitive descriptions as well.

Abstract:

“Ebinur Lake virus (EBIV) has been verified with highly virulent pathogenic to adult laboratory mice, and antibodies against EBIV have been detected in humans.” – please verify gramar mistakes, redundancy ...(EBIV) has been shown to be highly pathogenic... 

Does these antibodies were veirified by PRNT, to rule out cross reactions? I sugest to keep it simple and remove this part from abstract (...and antibodies against EBIV have been detected in humans).

It was showed that EBIV can be transmitted – how? Does the mosquito secrete virus particles in their saliva? How much in viral titers? Viral load? Did the authors performed experimental infection in mice with these experimentally infected mosquitoes? These points should be reflected in the abstract

Intrathoracic infection is not the natural infection route for vectors. Since mosquitoes feed on infected blood, oral feeding is the most likely route to reproduce what hapens in nature.

Introduction: the entire topic should be reformulated to be strait to the point. 

Line 50: are primarily distributed to three families and one order: Flaviviridae, Togaviridae and Reoviridae belong to orders Amarillovirales, Martellivirales and Reovirales. Sugestion only: use either the four orders and discriminate families, or use the names of the families within Bunyavirales were arboviruses are classified instead. 

Please use the current ICTV classification of viral families.

This part o f the introduction should be removed: 

Lines 56-60: “For example, the most notable flavivirus DENV, the pathogen of dengue 56 fever, has increased dramatically within the past 20 years, and more than 3.9 billion people in 128 countries are reported to be at risk of dengue infection [4]. After ZIKV and WNV Zika virus are introduced into the Western hemisphere, the viruses have a rapid geographical spread, and cause a large number of infections in the population [5,6].”

Introduction should be designed to what is really necessary to understand the importance of the study and the description of novel characteristics of this novel Peribunyaviridae.

Lines 64-66: “So far, the orthobunyaviruses can be discovered in mosquitoes of the different species, such as Ochlerotatus spp., Culex spp. and Aedes spp. [8-10].” Verify gramar mistakes, please: example: “So far, orthobunyaviruses have been descovered to infect different mosquito species...”

Line 67: Oropouche virus cause self-limiting acute febrile disease and, in 5% of infected people, mainly children, can cause asseptic meningitis.

Overall, this paragraph seems fragmented, needs a revision to be direct, concise, linked to the rest o f the introduction, considerations about flaviviruses, viruses circulating in Europe, should be entirely removed.

 It is necessary to explain what is known about this novel virus, and the importance of studying and discovering new viruses before they became a public health problem. 

Observe verb conjugation in this sentence: “Ebinur Lake virus (EBIV), a newly identified orthobunyavirus in China, is isolated from Culex modestus mosquito pools in Xinjiang Province” – replace “is” for “was” or “has been”.

Line 87-89: “EBIV can efficiently infect cells derived from rodent, avian, non-human primate, mosquito and human.” Were these tests performed in cell lines or primary cells? Please discriminate in the sentence.

Lines 89-90: “After the EBIV infection, BALB/c mice show the encephalopathy, hepatic damage and immunological system damages with high mortality” please verify gramar:

Sugestion - EBIV induced encephalopathy, hepatic and immunological system damages with high mortality index in experimentally infected BALB-c mice”

Lines 90-91: “Even though there is no report about confirmed human cases of EBIV, the serological proofs for EBIV infection has been detected”

Considering cross-reactions, please indicate wether this study was performed by PRNT or other serological test and introduce this possibility here.

Xia H, Liu R, Zhao L, Sun X, Zheng Z, Atoni E, et al. Characterization of Ebinur Lake Virus

636 and Its Human Seroprevalence at the China-Kazakhstan Border. Front Microbiol. 2019;10:3111: “Of the 211 tested serum samples, 17 were identified as IgM positive (1:4), 26 as IgG positive (1:10), and 4 as both IgM and IgG positive for EBIV by IFA (Supplementary Figure 3). In the participant with fever, higher IgM (10.37% vs. 0) and IgG (14.63 vs. 4.26%) positive rates were observed. Female participants had a much higher IgM-positive rate than male participants (13.95 vs. 4.00%), but no obvious difference was observed for IgG (13.95 vs. 9.6%). Study participants belonging to the >60-year age group had the highest positive rate of both IgM (16.67%) and IgG (12.69%) among all age groups. Furthermore, based on the occupational groups, the proportion of positive samples for IgM and IgG was highest among retirees (20%), followed closely by farmers and factory workers (17.28%). In addition, the neutralizing antibody prevalence of EBIV was 0.95% (2/211) (Table 1). For the two neutralizing antibody-positive cases, one male participant aged 50 years had a 1:8 PRNT90 titer, and the second case was from a female participant aged 50 years with a 1:16 PRNT90 titer. Both of these two study participants were from the Fifth Division of Xinjiang Production and Construction Corps region.”

Does the PRNT was performed only with EBIV? Did the test included other possible circulating orthobunyavirus to compare titers?

Line 92: Due to the health risk of EBIV: is there any clinical description in humans? If not, remove this sentence.

This virus was described in a Culex species. Culex sp. are more frequently associated to orthobunyaviruses than Aedes sp. Why the authors choosed Aedes aegypti instead of Culex mosquitoes colonies to perform this experimental study?

Lines 99-101: which is beneficial to better prepare for and respond to potential outbreak of EBIV, and to understand the transmission mechanism of orthobunyavriuses. – I strongly sugest this sentence to be removed, since by the results of previous studies, is not enoguh evidence of human infection. I also recommend that animals from the region should be tested for the virus.

Material and methods:

Line 111: “The EBIV isolate Cu20-XJ is obtained from Cx. modestus mosquitoes in 2013” – please verify verb conjugation (here is necessary to use past not present time).

Line 117: “Eggs of Ae. aegypti (Rockefeller 117 strain) was obtained from Laboratory of...” eggs – plural, was – singular.

Lines 120-121: ...”and a relative humidity of 75 ± 5% humidity, approximately.” Humidity is repeated.

Lines 125-126: ...”Then put the cups into the mesh cages (30 x 30 x 30 cm), which were kept in the insect incubator with the condition...” ...Cups were kept into the mesh....

Lines 132-133: Before oral-infection, five to eight days old, adult mosquitoes were collected by using the mosquito absorbing machine and placed into plastic cups (24oz). And wrap the cup with a cut mosquito net mesh and then cover it with a lid with a hole in the middle.

... “and placed into plastic cups (24oz), wrapped with a mosquito net mesh and covered with a lid containing a oppening in the center”...

Line 165: future tense should me avoided in manuscripts. Item 2.4 seems to be a “copy and paste” from Project arquive. Really hard to read.

Line 211: “Ten mosquitoes as a pool were used to extract total RNA using TRIzol reagent (Invitrogen).” ¬sugestion: “Each pool (n=10 especimens) was subjected to total RNA extraction using...”.

Line 226: “The clean reads were performed de novo assembly” did you mean: “Clean reads were used for de novo asembly?”

Results:

EBIV can be transmitted by Ae. aegypti through oral feeding – actually, in this manuscript transmission to a vertebrate was not evaluated. The most corrected presentation of this data is as the presence of EBIV titers in aedes aegypti saliva, or that EBIV infect several mosquito tissues.

In order to assess transmission rates, non infected mice should be exposed to these mosquitoes, and their tissues tested for the presence of the virus after the incubation period.

271-276: “The mean titers in the EBIV-positive guts at 4 (102.7 PFU ml-1), 7 (102.9 PFU ml-1), and 10 (102.8 PFU 272 ml-1) dpi were higher than those in heads, saliva and ovaries (Fig. 1E). At 14 dpi the viral titer in heads (103.5 PFU ml-1) was similar with the guts (103.2 PFU ml-1) at the same point (Fig. 1F). It was worth noting that the viral titer in saliva could get a highest virus titer at 14 dpi (102.5 PFU ml-1) (Fig. 1G) and at this time point ovaries also get the highest mean virus titer (103.6 PFU ml-1)” – By these titers, is not possible to afirm that aedes aegypti transmit the virus. Is necessary to expose a experimental model to these infected mosquitoes to prove that. 

Line 297: EBIV was highly susceptible in Ae. aegypti through intrathoracic inoculation – this objective should be more clear. By this title, it was inferred that the virus could not pass by mosquito gut barriers. However, Aedes aegypti was susceptible to virus infection, not otherwise, and by this route of inoculation, viral titers were higher in mosquito tissues than by oral route. 

It seems initially that intrathoracic route was used after oral route because viral titers or infection was not that efficient by the most similar route of infection in nature (oral feeding). In other places, it seems that this inoculation was used to evaluate mosquito innate response. Please review the results and abstract section to define these contents more properly.

“The mean titers in the EBIV-positive guts at 4 dpi (105.0 PFU ml−1) and 10 dpi (105.1 PFU ml−1) were significantly higher than that at 2 dpi (104.1 PFU ml−1, H = 13.70 and p = 0.0033) (Fig. 1E).” – and higher than in orally infected mosquitoes at these time points. 

Lines 338-339: “could get very high values” – I do not agree these are very hogh titers. These are medium values representing that the viruses efectively replicated in these tissues.

Lines 341-343: “With the increase of days after infection, the transmission rates were also slightly increased, ranging from 34.5-90%. The highest transmission rate...”

I do not think transmission rate is adequate to these results. Secretion/excretion rate, since transmission means actually the infection to be successful in a host exposed to these mosquitoes.

Discussion: overall, this is the most well written parto f the manuscript. Results are properly explored. I would like to know what the authors have to say about Ae. Aegypti been just a model, as the natural vector or reservoir might be a Culex species. Also, if the authors intend to perform mosquito experimental infection and test their transmission potential to experimental models (mice). Oral infection is the natural route of vector infection. Intrathoracic inoculation is not a natural route of infection. What is the explanation for the discrepant results between these two routes?

Reviewer #3: I have only a few major points for the authors to consider/clarify and a number of minor points that are easy to fix. Please see the "Summary and General Comments" section.

**Summary and General Comments**

Reviewer #1: Line 47 - Mosquito-borne viruses (MBVs), as a group of heterogeneous RNA viruses, naturally survive in both mosquitoes and vertebrate hosts, and are the aetiological agents of many human diseases. 

"Mosquito-borne viruses" may be confusing because of the insect-specific viruses that do not replicate in vertebrate cells. It seems that Ebinur Lake virus has demonstrated capacity to infect vertebrate and invertebrate cell lines so it can be categorized as an arbovirus, which is a term well-established. Also, viruses replicate rather than survive.

I suggest changing to:

Arthopod-borne viruses (arbovirus), as a group of heterogeneous RNA viruses, naturally replicate in both mosquitoes and vertebrate hosts, and are the aetiological agents of many human diseases.

Line 49 - The medically important MBVs are primarily distributed to three families and one order: the Flaviviridae family [dengue viruses 1-4 (DENV), Zika virus (ZIKV), West Nile virus (WNV), Japanese encephalitis virus (JEV), etc], the Togaviridae family [Chikungunya virus (CHIKV), etc], the Reoviridae order [(Banna virus (BAV), etc) and Bunyavirales order [Rift Valley fever virus (RVFV), etc] [2, 3].

I suggest changing to:

Line 49 - The medically important arboviruses are primarily distributed to four families: Flaviviridae [dengue viruses 1-4 (DENV), Zika virus (ZIKV), West Nile virus (WNV), Japanese encephalitis virus (JEV), etc], Togaviridae [Chikungunya virus (CHIKV), Mayaro virus (MAYV) etc], Reoviridae [Banna virus (BAV), etc], and Peribunyaviridae [Oropouche orthobunyavirus (OROV) etc] [2, 3].

Line 57 - After ZIKV and WNV Zika virus are introduced into the Western hemisphere, the viruses have a rapid geographical spread, and cause a large number of infections in the population

I suggest changing to:

After ZIKV and WNV were detected in the Western hemisphere, they rapidly spread causing large number of neurological disorders mainly in Brazil and United States, respectively.

Line 60 - change "mosquito-borne" by "arbovirus", and add the proper references.

Line 65 - change "Ochlerotatus spp., Culex spp. and Aedes spp. [8-10]" by "Ochlerotatus spp., Culex spp. and Aedes spp. and also midges, as Culicoides paraensis, which is a vector of OROV in South America [8-10]" 

From line 92 to 95 - Authors should indicate here why not using Culex spp for the experimental infection, or at least describe if any experimental work has been done with Culex spp. EBIV was detected in Culex modestus, so it would make sense evaluate the vector capacity and competence of the most common Culex spp. in China as well. This topic also needs to be discussed in the discussion section.

Line 117 - Authors need to explain and discuss at some point in the manuscript the advantages and disadvantages of using the Rockefeller strain of Ae. aegypti for these experiments. We know that different populations of Ae. aegypti can have different vectorial capacity for other arboviruses, so it would be helpful for the readers to know why not using a wild population of Ae. aegypti reared from eggs collected in the field in China. Wild populations would reflect a more realistic vectorial capacity due ecological and biological characteristics that could interfere with EBIV replication, as the infection by insect-specific viruses etc. Perhaps this can be considered a first step for the vector competence analysis, but it needs t be clear for the reader the reasons for the chosen approach.

Line 136 - Mosquitoes were fed with supernatant of EBIV-infected BHK cells

Line 143 - Under the dissecting microscope, insert the loaded needle (filled with supernatant of EBIV-infected BHK cells)

I noticed the virus concentration used for inoculation and blood meal is described in the results section, but I wonder if it would not be easier for the reader having that information here at the material and methods.

Line 147 - "nl of virus" My understanding is that authors did not concentrate virus for inoculation, so here "virus" needs to be replaced by supernatant of EBIV-infected BHK cells.

Line 186 - "mice anti-EBIV-NP" Was this primary antibody prepared by the authors in previous study? If so, cite here. Is this mouse hyper immune ascitic fluid? If so, provide that information here.

Line 248 to 252 could be transferred to the material and methods section. 

Line 252 to 256 - It is not very clear if titration was done by the equation described on line 177 and 178 or plaque assay. If by the described equation, please explain it better, citing references that support the calculation used. If plaque assay was used, describe the method at some point at material and methods.

Line 381-384 - "we compared the differences of gene expression between the intrathoracic inoculation and

mock-injected mosquitoes at the same time point". Please, make clear here, or later in line 470-473 in the discussion section, the reason for not using orally-infected mosquitoes to evaluate down-regulation of immune-related genes.

Lines 468-469 and 482-484 describe the same potential for vertical transmission.

Discussion section - Overall, I believe the discussion section can be improved. I missed in the discussion section the limitations of this study, which could enrich the discussion of this laborious study. Authors could discuss the potential different responses using different field Ae. aegypti populations, the experimental infection with Culex spp, etc. Also, the increasing followed by reduction in dissemination and infection rates observed at 10 dpi etc. Discussion of down-regulation of genes, for instance, is restricted to a couple of lines and should be better explored.

Reviewer #2: Overall, the main goal of this study is to demonstrate that viral replication alters gene expression in an experimental vector model. Therefore, after an extensive review, i recommend this article to be evaluated again by reviewers.

Reviewer #3: This manuscript describes the EBIV- Aedes aegypti interactions at the vector competence level. Two infection ways possess divergent infection, dissemination rate, and ovary infection rates at different dpi. The authors also present the morphology of EBIV in different organs via IFA and EM. Finally, the transcriptome data provide the relevant immune/ metabolism genes expression profile, potentially associated with the vector transmission capability. The work is well-executed, with an adequate number of replicates and controls. The study has generated novel results of interest and increased our knowledge on this newly classified mosquito-borne orthobunyavirus and the vector. I have only a few major points for the authors to consider/clarify and a number of minor points that are easy to fix.

1-I wonder if the title should be more specific to better reflect the results obtained in this work.

2- The English needs to be clarified so that the reader clearly understands what is being said. This must be done throughout the document, but here are some examples:

L21. ... diseases has been increasing in the last...

L25. ... it is necessary to assess mosquitoes' vector capacity for EBIV to predict its risk to...

L27... was shown that EBIV could be...

L33. Delete “all”

L33. …demonstrated EBIV could alter…

L34. processes

3-I have some comments on the methods. The authors need to be clarified.

a. L112, please provide the information for mice, strain, age.

b. L114, a reference, needs to cite to describe this plaque assay.

c. L113, adult female?

d. It would be better to provide the numbers of mosquitoes in each experiment, such as viral detection, IFC, and EM.

e. a reference need to cite for the rates used in this manuscript.

f. filter information for confocal.

4- I have the feeling that this 3.7 of the Results section is some kind of last minute add-on. I would suggest adding more words in the relevant pathways, especially for the functions in the modulation of virus infection.

5- I favor future work, and possible points would be investigated in the Discussion section, but some words in the discussion need to be said for the transcriptome data. I have some suggestions for the authors to consider. For example, the function of the up-regulation genes in immune response; how the metabolism pathways work on the virus infection?

PLOS authors have the option to publish the peer review history of their article (what does this mean?). If published, this will include your full peer review and any attached files.

Reviewer #1: No

Reviewer #2: No

Reviewer #3: Yes: Xin-Ru Wang
---

## [Decision Letter · Decision Letter 1]

21 Jun 2022

Dear Dr. Xia,

Thank you very much for submitting your manuscript "Vector competence and immune response of Aedes aegypti for Ebinur Lake virus, a newly classified mosquito-borne orthobunyavirus" for consideration at PLOS Neglected Tropical Diseases. As with all papers reviewed by the journal, your manuscript was reviewed by members of the editorial board and by several independent reviewers. The reviewers appreciated the attention to an important topic. Based on the reviews, we are likely to accept this manuscript for publication, providing that you modify the manuscript according to the review recommendations. 

Sincerely,

Adly M.M. Abd-Alla, Prof asso.

Associate Editor

Eric Dumonteil

Deputy Editor

Reviewer's Responses to Questions

**Key Review Criteria Required for Acceptance?**

**Methods**

-Are the objectives of the study clearly articulated with a clear testable hypothesis stated?

-Is the study design appropriate to address the stated objectives?

-Is the population clearly described and appropriate for the hypothesis being tested?

-Is the sample size sufficient to ensure adequate power to address the hypothesis being tested?

-Were correct statistical analysis used to support conclusions?

-Are there concerns about ethical or regulatory requirements being met?

Reviewer #2: objectives are clearly stated, study design seems to be firstly made to prove oral natural infection occur in this species and virus reach salivary glands. In a second moment, the authors choosed intrathoracic inoculation to overcome gut barrier of virus dissemination to stablish persistent salivary gland infection and study which genes are modulated by virus replication in mosquitoes (Ae aegypti). I am not convinced by the results that Ae aegypti is a good choice, either that this species could sustain virus transmission of this virus. 

sample size is ok

ethical or regulatory requirements are ok too.

Reviewer #3: (No Response)

**Results**

-Does the analysis presented match the analysis plan?

-Are the results clearly and completely presented?

-Are the figures (Tables, Images) of sufficient quality for clarity?

Reviewer #2: Current version is a lot more clear and in a better fashion.

-The analysis presented match the analysis plan? Abstract contains few affirmations not sustained by the presented results that should be reviewed again. Ovary infection rates and saliva titers after intrathoracic infection, and gene expression modulation by viral infection are important results of the study. 

-Are the results clearly and completely presented? See comments above.

-Are the figures (Tables, Images) of sufficient quality for clarity? Yes. Figures are very clear.

Reviewer #3: (No Response)

**Conclusions**

-Are the conclusions supported by the data presented?

-Are the limitations of analysis clearly described?

-Do the authors discuss how these data can be helpful to advance our understanding of the topic under study?

-Is public health relevance addressed?

Reviewer #2: -Are the conclusions supported by the data presented? See considerations about abstract. Discussion is a lot improved.

-Are the limitations of analysis clearly described? It is a lot improved in this version.

-Do the authors discuss how these data can be helpful to advance our understanding of the topic under study? Yes.

-Is public health relevance addressed? These studies do not implicate public health directly. Few data relate this pathogen to humans in literature. This study contributes to understand Ae aegypti replication of the virus. Main findings are related to mosquito model, mosquito immune response activation, importance of gut barrier to limit virus dissemination in mosquito body to reach salivary glands, possible mosquito vertical transmission since the virus reached ovaries. Biological importance in an anthropophilic mosquito species is well demonstrated.

Reviewer #3: (No Response)

**Editorial and Data Presentation Modifications?**

Reviewer #2: (No Response)

Reviewer #3: (No Response)

**Summary and General Comments**

Reviewer #2: Main consideration:

This version of manuscript is a lot more uniform, direct and concise. Importance of invertebrate model studies is clearly demonstrated by the findings, since EBIV infected ovary tissues (reflecting it possibly is transmitted to offsprings) and viral titers are present in saliva. Also, demonstrate gut barrier limited virus dissemination after natural infection and, virus replication leaded to gene expression related to mosquito immune response. These findings are well demonstrated and should be evidenced in the abstract. Therefore, abstract should be reformulated to properly define results and findings of the study. 

Public health importance should be minimized since i. human transmission was not accessed in this manuscript; ii. the study did not show a high frequency of positivity in the saliva of Ae aegypti when infection is made with high viral titer through oral route, only when non-natural, intrathoracic route is used. Gut barrier probably limited the virus dissemination to salivary glands, leading to few mosquitoes with virus in their saliva after oral infection, when considering the total number of mosquitoes included in the experiment/evaluated in each time period (figure 1G). Therefore, this observations should be tacking into consideration to complete review of the manuscript.

By the results, is not possible to say this species could spread the virus, since a high number of mosquitoes infected through oral route did not show virus titers in saliva.

Should reflect these observations:

“Following intrathoracic injection, EBIV bypassed the midgut barrier and was able to rapidly infect the whole mosquito body, including saliva and ovary (Fig. 2G to H). The distinct results obtained with the two routes indicate that the midgut of Ae. aegypti is the main barrier to EBIV transmission.”

In my opinion, oral infection does not seem to be very efficient in this species of mosquito, next studies should consider to use other mosquito population.

Seems gut barrier limits the establishment of persistent infection in salivary glands of Ae aegypti even when a high viral load is inoculated. Only a few mosquitoes of this species are infected through natural vector infection route, considering the number of mosquitoes included in the experiment. 

Therefore, implications to public health at this point, should not be made.

Author summary is a lot better focused in the biological importance of the main findings of the study. Abstract should be reformulated to continuous that line of thinking.

Lines 94-95: “it poses a considerable risk to public health and may offer a new threat to human or animal health and economic prosperity” –is not enough data to affirm this virus is a public health threat. In my opinion, the study should exclude these affirmations and reflect the important findings and conclusions related to mosquitoes. As far as is possible to comprehend, this virus is important to animal health, not yet proved with sufficient data that it can cause an outbreak involving humans.

Discussion section should be reflected on abstract writing.

Reviewer #3: (No Response)

PLOS authors have the option to publish the peer review history of their article (what does this mean?). If published, this will include your full peer review and any attached files.

Reviewer #2: No

Reviewer #3: No

Figure Files:

Data Requirements:

Reproducibility:

References

---

## [Editor Report · Decision Letter 2]

8 Jul 2022

Dear Dr. Xia,

We are pleased to inform you that your manuscript 'Vector competence and immune response of Aedes aegypti for Ebinur Lake virus, a newly classified mosquito-borne orthobunyavirus' has been provisionally accepted for publication in PLOS Neglected Tropical Diseases.

Best regards,

Adly M.M. Abd-Alla, Prof asso.

Associate Editor

Eric Dumonteil

Deputy Editor

---

## [Editor Report · Acceptance letter]

14 Jul 2022

Dear Dr. Xia,

We are delighted to inform you that your manuscript, "Vector competence and immune response of *Aedes aegypti* for Ebinur Lake virus, a newly classified mosquito-borne orthobunyavirus ," has been formally accepted for publication in PLOS Neglected Tropical Diseases.

Best regards,

Shaden Kamhawi

co-Editor-in-Chief

Paul Brindley

co-Editor-in-Chief
